# A Case Report: Multifocal Necrotizing Encephalitis and Myocarditis after BNT162b2 mRNA Vaccination against COVID-19

**DOI:** 10.3390/vaccines10101651

**Published:** 2022-10-01

**Authors:** Michael Mörz

**Affiliations:** Institute of Pathology ’Georg Schmorl’, The Municipal Hospital Dresden-Friedrichstadt, Friedrichstrasse 41, 01067 Dresden, Germany; michael.moerz@klinikum-dresden.de

**Keywords:** COVID-19 vaccination, necrotizing encephalitis, myocarditis, detection of spike protein, detection of nucleocapsid protein, autopsy

## Abstract

The current report presents the case of a 76-year-old man with Parkinson’s disease (PD) who died three weeks after receiving his third COVID-19 vaccination. The patient was first vaccinated in May 2021 with the ChAdOx1 nCov-19 vector vaccine, followed by two doses of the BNT162b2 mRNA vaccine in July and December 2021. The family of the deceased requested an autopsy due to ambiguous clinical signs before death. PD was confirmed by post-mortem examinations. Furthermore, signs of aspiration pneumonia and systemic arteriosclerosis were evident. However, histopathological analyses of the brain uncovered previously unsuspected findings, including acute vasculitis (predominantly lymphocytic) as well as multifocal necrotizing encephalitis of unknown etiology with pronounced inflammation including glial and lymphocytic reaction. In the heart, signs of chronic cardiomyopathy as well as mild acute lympho-histiocytic myocarditis and vasculitis were present. Although there was no history of COVID-19 for this patient, immunohistochemistry for SARS-CoV-2 antigens (spike and nucleocapsid proteins) was performed. Surprisingly, only spike protein but no nucleocapsid protein could be detected within the foci of inflammation in both the brain and the heart, particularly in the endothelial cells of small blood vessels. Since no nucleocapsid protein could be detected, the presence of spike protein must be ascribed to vaccination rather than to viral infection. The findings corroborate previous reports of encephalitis and myocarditis caused by gene-based COVID-19 vaccines.

## 1. Introduction

The emergence of the severe acute respiratory syndrome coronavirus 2 (SARS-CoV-2) in 2019 with the subsequent worldwide spread of COVID-19 gave rise to a perceived need for halting the progress of the COVID-19 pandemic through the rapid development and deployment of vaccines. Recent advances in genomics facilitated gene-based strategies for creating these novel vaccines, including DNA-based nonreplicating viral vectors, and mRNA-based vaccines, which were furthermore developed on an aggressively shortened timeline [1,2,3,4].

The WHO Emergency Use Listing Procedure (EUL), which determines the acceptability of medicinal products based on evidence of quality, safety, efficacy, and performance [5], permitted these vaccines to be marketed as soon as 1–2 years after development had begun. Published results of the phase 3 clinical trials described only a few severe side effects [2,6,7,8]. However, it has since become clear that severe and even fatal adverse events may occur; these include in particular cardiovascular and neurological manifestations [9,10,11,12,13]. Clinicians should take note of such case reports for the sake of early detection and management of such adverse events among their patients. In addition, a thorough post-mortem examination of deaths in connection with COVID-19 vaccination should be considered in ambiguous circumstances, including histology. This report presents the case of a senior aged 76 years old, who had received three doses overall of two different COVID-19 vaccines, and who died three weeks after the second dose of the mRNA-BNT162b-vaccine. Autopsy and histology revealed unexpected necrotizing encephalitis and mild myocarditis with pathological changes in small blood vessels. A causal connection of these findings to the preceding COVID-19 vaccination was established by immunohistochemical demonstration of SARS-CoV-2 spike protein. The methodology introduced in this study should be useful for distinguishing between causation by COVID-19 vaccination or infection in ambiguous cases.

## 2. Materials and Methods

### 2.1. Routine Histology 

Formalin-fixed tissues were routinely processsed and paraffin-embedded tissues were cut into 5 μm sections and stained with hematoxylin and eosin (H&E) for histopathological examination.

### 2.2. Immunohistochemistry

Immunohistochemical staining was performed on the heart and brain, using a fully automated immunostaining system (Ventana Benchmark, Roche). An antigen retrieval (Ultra CC1, Roche Ventana) was used for every antibody. The target antigens and dilution factors for the antibodies used are summarized in Table 1. Incubation with the primary antibody was carried out for 30 min in each case. Tissues from SARS-CoV-2-positive COVID-19 patients were used as a control for the antibodies against SARS-CoV-2-spike and nucleocapsid (Figure 1). Cultured cells that had been transfected in vitro (see hereafter) served as a positive control for the detection of vaccine-induced spike protein expression and as a negative control for the detection of nucleocapsid protein. The slides were examined with a light microscope (Nikon ECLIPSE 80i) and representative images were captured by the camera system Motic^®^ Europe Motic MP3. 

### 2.3. Preparation of Positive Control Samples for the Immunohistochemical Detection of the Vaccine-Induced Spike Protein

Cell culture and transfection: Ovarian cancer cell lines (OVCAR-3 and SK-OV3, CSL cell Lines Service, Heidelberg, Germany) were grown to 70% confluence in flat bottom 75 cm^2^ cell culture flasks (Cell star) in DMEM/HAMS-F12 medium supplemented with Glutamax (Sigma-Aldrich, St. Louis, MO, USA), 10% FCS (Gibco, Shanghai, China) and Gentamycin (final concentration 20 μg/mL, Gibco), at 37 °C, 5% CO_2_ in a humidified cell incubator. For transfection, the medium was completely removed, and cells were incubated for 1 h with 2 mL of fresh medium containing the injection solutions directly from the original bottles, diluted 1:500 in the case of BNT162b2 (Pfizer/Biotech), and 1:100 in cases of mRNA-1273 (Moderna), Vaxzevria (AstraZeneca), and Jansen (COVID-19 vaccine Jansen). Then, another 15 mL of fresh medium was added to the cell cultures and cells were grown to confluence for another 3 days. 

Preparation of tissue blocks from transfected cells: The cell culture medium was removed from transfected cells, and the monolayer was washed twice with PBS, then trypsinized by adding 1 mL of 0.25% Trypsin-EDTA (Gibco), harvested with 10 mL of PBS/10% FCS, and washed 2× with PBS and centrifugation at 280× *g* for 10 min each. Cell pellets were fixed overnight in 2 mL in PBS/4% Formalin at 8 °C and then washed in PBS once. The cell pellets remaining after centrifugation were suspended in 200 μL PBS each, mixed with 400 μL 2% agarose in PBS solution (precooled to around 40 °C), and immediately transferred to small (1 cm) dishes for fixation. The fixed and agarose-embedded cell pellets were stored in 4% Formalin/PBS till subjection to routine paraffin embedding in parallel to tissue samples.

### 2.4. Case Presentation and Description

#### 2.4.1. Clinical History

This report presents the case of a 76-year-old male with a history of Parkinson’s disease (PD) who passed away three weeks after his third COVID-19 vaccination. On the day of his first vaccination in May 2021 (ChAdOx1 nCov-19 vector vaccine), he experienced pronounced cardiovascular side effects, for which he repeatedly had to consult his doctor. After the second vaccination in July 2021 (BNT162b2 mRNA vaccine/Comirnaty), the family noted obvious behavioral and psychological changes (e.g., he did not want to be touched anymore and experienced increased anxiety, lethargy, and social withdrawal even from close family members). Furthermore, there was a striking worsening of his PD symptoms, which led to severe motor impairment and a recurrent need for wheelchair support. He never fully recovered from these side effects after the first two vaccinations but still got another vaccination in December 2021. Two weeks after the third vaccination (second vaccination with BNT162b2), he suddenly collapsed while taking his dinner. Remarkably, he did not show coughing or any signs of food aspiration but just fell down silently. He recovered from this more or less, but one week later, he again suddenly collapsed silently while taking his meal. The emergency unit was called, and after successful, but prolonged resuscitation attempts (over one hour), he was transferred to the hospital and directly put into an artificial coma but died shortly thereafter. The clinical diagnosis was death due to aspiration pneumonia. According to his family, there was no history of a clinical or laboratory diagnosis of COVID-19 in the past.

#### 2.4.2. Autopsy

The autopsy was requested and consented to by the family of the patient because of the ambiguity of symptoms before his death. The autopsy was performed according to standard procedures including macroscopic and microscopic investigation. Gross brain tissue was prepared for histological examination including the brain (frontal cortex, Substantia nigra, and Nucleus ruber) as well as the heart (left and right ventricular cardiac tissue).

## 3. Results

### 3.1. Autopsy Findings

Anatomical Specifications: Body weight, height, and specifications of body organs were summarized in Table 2.

Brain: A macroscopic examination of brain tissue revealed a circumscribed segmental cerebral parenchymal necrosis at the site of the right hippocampus. Substantia nigra showed a loss of pigmented neurons. Microscopically, several areas with lacunar necrosis were detected with inflammatory debris reaction on the left frontal side (Figure 2). Staining of Nucleus ruber with H&E showed neuronal cell death, microglia, and lymphocyte infiltration (Figure 3). Furthermore, there were microglial and lymphocytic reactions as well as predominantly lymphocytic vasculitis, sometimes with mixed infiltrates including neutrophilic granulocytes (Figure 4) in the frontal cortex, paraventricular, Substantia nigra, and Nucleus ruber on both sides. In some places with inflammatory changes in brain capillaries, there were also signs of apoptotic cell death within the endothelium (Figure 4). Meninges’ findings were unremarkable. The collective findings were suggestive of multifocal necrotizing encephalitis. Furthermore, chronic arteriosclerotic lesions of varying degrees were noted in large brain vessels, which are described in detail in section “Vascular system”.

Parkinson’s disease (PD): Macroscopic and histological examination of brain tissue revealed bilateral pallor of the substantia nigra with loss of pigmented neurons. In addition, pigment-storing macrophages as well as scattered neuronal necrosis with glial debris reaction were noted. These findings were suggestive of PD, confirming the clinical diagnosis.

Thoracic cavity: An examination of the chest showed a funnel-shaped chest with serial rib fractures (extending from the second to fifth ribs on the right, and from the second to sixth ribs on the left); which is a common picture of a patient who underwent cardiopulmonary resuscitation. An endotracheal tube was properly inserted. There was evidence of regular placement of a central venous catheter in the left femoral vein. There was evidence of regular placement of an arterial catheter in the left radial artery. The urinary catheter was inserted as well. There was a 9 cm long skin scar on the front of the right shoulder.

Lungs: Macroscopical lung examination revealed cloudy secretion and purulent spots with notably brittle parenchyma. The pleura showed bilateral serous effusion, amounting to 450 mL of fluid on the right side and 400 mL on the left side. Bilateral mucopurulent tracheobronchitis was evident with copious purulent secretion in the trachea and bronchi. Bilateral chronic destructive pulmonary emphysema was detected. Bilateral bronchopneumonia was noted in the lower lung lobes at multiple stages of development and lobe-filling with secretions and fragile parenchyma. Furthermore, chronic arteriosclerotic lesions of varying degrees were noted, which are described in detail in the section “Vascular system”.

Heart: Macroscopic cardiac examination revealed manifestations of acute and chronic cardiovascular insufficiency, including ectasia of the atria and ventricles. Furthermore, left ventricular hypertrophy was noted (wall thickness: 18 mm, heart weight: 410 g, body weight: 60 kg, height: 1.75 m). There was evidence of tissue congestion (presumably due to cardiac insufficiency) in the form of pulmonary edema, cerebral edema, brain congestion, chronic hepatic congestion, renal tissue edema, and pituitary tissue edema. Moreover, there was evidence of shock kidney disorder. Histological examination of the heart revealed mild myocarditis with fine-spotted fibrosis and lympho-histiocytic infiltration (Figure 5). Furthermore, there were chronic arteriosclerotic lesions of varying degrees, which are described in detail under “Vascular system”. In addition to these, there were more acute myocardial and vascular changes in the heart. They consisted of mild signs of myocarditis, characterized by infiltrations with foamy histiocytes and lymphocytes as well as hypereosinophilia and some hypercontraction of cardiomyocytes. Furthermore, mild acute vascular changes were observed in the capillaries and other small blood vessels of the heart. They consisted of mild lympho-histiocytic infiltrates, prominent endothelial swelling and vacuolation, multifocal myocytic degeneration and coagulation necrosis as well as karyopyknosis of single endothelial cells and vascular muscle cells (Figure 5). Occasionally, adhering plasma coagulates/fibrin clots were present on the endothelial surface, indicative of endothelial damage (Figure 5).

Vascular system (large blood vessels): The pulmonary arteries showed ectasia and lipidosis. The kidney showed slight diffuse glomerulosclerosis and arteriosclerosis with renal cortical scars (up to 10 mm in diameter). The findings are suggestive of generalized atherosclerosis and systemic hypertension. Major arteries including the aorta and its branches as well as the coronary arteries showed variable degrees of arteriosclerosis and mild to moderate stenosis. Furthermore, examination revealed mild nodular arteriosclerosis of cervical arteries. Ascending aorta, aortic arch, and thoracic aorta showed moderate, nodular, and partially calcified arteriosclerosis. The cerebral basilar artery showed mild arteriosclerosis. Nodular and calcified arteriosclerosis were of high grade in the abdominal aorta and iliac arteries and moderate grade with moderate stenosis in the right coronary arteries. Coronary artery examination showed variable degrees of arteriosclerosis and stenosis more on the left coronary arteries. The left anterior descending coronary artery (the anterior interventricular branch of the left coronary artery; LAD) showed high-grade and moderately stenosed arteriosclerosis. The arteriosclerosis and stenosis of the left circumflex artery (the circumflex branch of the left coronary artery) were mild. Mild cerebral basal artery sclerosis. High-grade nodular and calcified arteriosclerosis of the abdominal aorta and the iliac arteries. Moderate stenosed arteriosclerosis of the right coronary artery. Lymphocytic periarteritis was detected as well.

### 3.2. Other Findings

-Oral cavity: tongue bite was detected with bleeding under the tongue muscle (tongue bite is common with epileptic seizures).-Adrenal glands: bilateral mild cortical hyperplasia.-Colon: the elongated sigmoid colon was elongated with fecal impaction.-Kidneys: slight diffuse glomerulosclerosis and arterio-sclerosis, renal cortical scars (up to 10 mm in diameter), bilateral mild active nephritis and urocystitis as well as evidence of shock kidney disorder.-Liver: slight lipofuscinosis.-Spleen: mild acute splenitis.-Stomach: mild diffuse gastric mucosal bleeding.-Thyroid gland: bilateral nodular goiter with chocolate cysts (up to 0.5 cm in diameter).-Prostate gland: benign nodular prostatic hyperplasia and chronic persistent prostatitis.

### 3.3. Immunohistochemical Analyses

Immunohistochemical staining for the presence of SARS-CoV-2 antigens (spike protein and nucleocapsid) was studied in the brain and heart. In the brain, SARS-CoV-2 spike protein subunit 1 was detected in the endothelia, microglia, and astrocytes in the necrotic areas (Figure 6 and Figure 7). Furthermore, spike protein could be demonstrated in the areas of lymphocytic periarteritis, present in the thoracic and abdominal aorta and iliac branches, as well as a cerebral basal artery (Figure 8). The SARS-CoV-2 subunit 1 was found in macrophages and in the cells of the vessel wall, in particular the endothelium (Figure 9), as well as in the Nucleus ruber (Figure 10). In contrast, the nucleocapsid protein of SARS-CoV-2 could not be detected in any of the corresponding tissue sections (Figure 11 and Figure 12). In addition, SARS-CoV-2 spike protein subunit 1 was detected in the cardiac endothelial cells that showed lymphocytic myocarditis (Figure 13). Immunohistochemical staining did not detect the SARS-CoV-2 nucleocapsid protein (Figure 14).

### 3.4. Autopsy-Based Diagnosis

The 76-year-old deceased male patient had PD, which corresponded to typical post-mortem findings. The main cause of death was recurrent aspiration pneumonia. In addition, necrotizing encephalitis and vasculitis were considered to be major contributors to death. Furthermore, there was mild lympho-histiocytic myocarditis with fine-spotted myocardial fibrosis as well as systemic arteriosclerosis, which will have also contributed to the deterioration of the physical condition of the senior.

The final diagnosis was abscedating bilateral bronchopneumonia (J18.9), Parkinson’s disease (G20.9), necrotic encephalitis (G04.9), and myocarditis (I40.9).

Immunohistochemistry for SARS-CoV-2 antigens (spike protein and nucleocapsid) revealed that the lesions with necrotizing encephalitis as well as the acute inflammatory changes in the small blood vessels (brain and heart) were associated with abundant deposits of the spike protein SARS-CoV-2 subunit 1. Since the nucleocapsid protein of SARS-CoV-2 was consistently absent, it must be assumed that the presence of spike protein in affected tissues was not due to an infection with SARS-CoV-2 but rather to the transfection of the tissues by the gene-based COVID-19-vaccines. Importantly, spike protein could be only demonstrated in the areas with acute inflammatory reactions (brain, heart, and small blood vessels), in particular in endothelial cells, microglia, and astrocytes. This is strongly suggestive that the spike protein may have played at least a contributing role to the development of the lesions and the course of the disease in this patient.

## 4. Discussion

This is a case report of a 76-year-old patient with Parkinson’s disease (PD) who died three weeks after his third COVID-19 vaccination. The stated cause of death appeared to be a recurrent attack of aspiration pneumonia, which is indeed common in PD [14,15]. However, the detailed autopsy study revealed additional pathology, in particular necrotizing encephalitis and myocarditis. While the histopathological signs of myocarditis were comparatively mild, the encephalitis had resulted in significant multifocal necrosis and may well have contributed to the fatal outcome. Encephalitis often causes epileptic seizures, and the tongue bite found at the autopsy suggests that it had done so in this case. Several other cases of COVID-19 vaccine-associated encephalitis with status epilepticus have appeared previously [16,17,18].

The clinical history of the current case showed some remarkable events in correlation to his COVID-19 vaccinations. Already on the day of his first vaccination in May 2021 (ChAdOx1 nCov-19 vector vaccine), he experienced cardiovascular symptoms, which needed medical care and from which he recovered only slowly. After the second vaccination in July 2021 (BNT162b2 mRNA vaccine), the family recognized remarkable behavioral and psychological changes and a sudden onset of marked progression of his PD symptoms, which led to severe motor impairment and recurrent need for wheelchair support. He never fully recovered from this but still was again vaccinated in December 2021. Two weeks after this third vaccination (second vaccination with BNT162b2), he suddenly collapsed while taking his dinner. Remarkably, he did not show any coughing or other signs of food aspiration but just fell from his chair. This raises the question of whether this sudden collapse was really due to aspiration pneumonia. After intense resuscitation, he recovered from this more or less, but one week later, he again suddenly collapsed silently while taking his meal. After successful but prolonged resuscitation attempts, he was transferred to the hospital and directly set into an artificial coma but died shortly thereafter. The clinical diagnosis was death due to aspiration pneumonia. Due to his ambiguous symptoms after the COVID-vaccinations the family asked for an autopsy.

Based on the alteration pattern in the brain and heart, it appeared that the small blood vessels were especially affected, in particular, the endothelium. Endothelial dysfunction is known to be highly involved in organ dysfunction during viral infections, as it induces a pro-coagulant state, microvascular leak, and organ ischemia [19,20]. This is also the case for severe SARS-CoV-2 infections, where a systemic exposure to the virus and its spike protein elicits a strong immunological reaction in which the endothelial cells play a crucial role, leading to vascular dysfunction, immune-thrombosis, and inflammation [21].

Although there was no history of COVID-19 for this patient, immunohistochemistry for SARS-CoV-2 antigens (spike and nucleocapsid proteins) was performed. Spike protein could be indeed demonstrated in the areas of acute inflammation in the brain (particularly within the capillary endothelium) and the small blood vessels of the heart. Remarkably, however, the nucleocapsid was uniformly absent. During an infection with the virus, both proteins should be expressed and detected together. On the other hand, the gene-based COVID-19 vaccines encode only the spike protein and therefore, the presence of spike protein only (but no nucleocapsid protein) in the heart and brain of the current case can be attributed to vaccination rather than to infection. This agrees with the patient’s history, which includes three vaccine injections, the third one just 3 weeks before his death, but no positive laboratory or clinical diagnosis of the infection.

Discrimination of vaccination response from natural infection is an important question and had been addressed already in clinical immunology, where the combined application of anti-spike and anti-nucleocapsid protein-based serology was proven as a useful tool [22]. In histology, however, this immunohistochemical approach has not yet been described, but it is straightforward and appears to be very useful for identifying the potential origin of SARS-CoV-2 spike protein in autopsy or biopsy samples. Where additional confirmation is required, for instance in a forensic context, rt-PCR methods might be used to ascertain the presence of the vaccine mRNA in the affected tissues [23,24].

Assuming that, in the current case, the presence of spike protein was indeed driven by the gene-based vaccine, then the question arises whether this was also the cause the accompanying acute tissue alterations and inflammation. The stated purpose of the gene-based vaccines is to induce an immune response against the spike protein. Such an immune response will, however, not only results in antibody formation against the spike protein but also lead to direct cell- and antibody-mediated cytotoxicity against the cells expressing this foreign antigen. In addition, there are indications that the spike protein on its own can elicit distinct toxicity, in particular, on pericytes and endothelial cells of blood vessels [25,26].

While it is widely held that spike protein expression, and the ensuing cell and tissue damage will be limited to the injection site, several studies have found the vaccine mRNA and/or the spike protein encoded by it at a considerable distance from the injection site for up to three months after the injection [23,24,27,28,29]. Biodistribution studies in rats with the mRNA-COVID-19 vaccine BNT162b2 also showed that the vaccine does not stay at the injection site but is distributed to all tissues and organs, including the brain [30]. After the worldwide roll-out of COVID-19 vaccinations in humans, spike protein has been detected in humans as well in several tissues distant from the injection site (deltoid muscle): for instance in heart muscle biopsies from myocarditis patients [28], within the skeletal muscle of a patient with myositis [23] and within the skin, where it was associated with a sudden onset of Herpes zoster lesions after mRNA-COVID-19 vaccination [29].

The underlying diagnosis in this patient was Parkinson’s disease, and one may ask what role, if any, this condition had played in the causation of the encephalitis, and the myocarditis detected at post-mortem examination. PD had been long-standing in the current case, whereas the encephalitis was acute. Conversely, there is no plausible mechanism and no case report of PD causing secondary necrotizing encephalitis. On the other hand, numerous cases have been reported of autoimmune encephalitis and encephalomyelitis after COVID-19 vaccination [12,31]. Autoimmune diseases in organs other than the CNS have been reported as well, for example, a striking case of a patient who after mRNA vaccination suffered multiple autoimmune disorders all at once—acute disseminated encephalomyelitis, myasthenia gravis, and thyroiditis [32]. In the case reported here, it may be noted that the spike protein was primarily detected in the vascular endothelium and sparsely in the glial cells but not in the neurons. Nevertheless, neuronal cell death was widespread in the encephalitic foci, which suggests some contribution of immunological bystander activation, i.e., autoimmunity, to the observed cell and tissue damage.

A contributory role of PD in the development of cardiomyopathy is indeed documented and cannot be ruled out with absolute certainty. However, inflammatory myocardial changes with pathological alterations in small blood vessels as seen in the current case are uncommon. Instead, the most prominent cause of cardiac failure in PD patients is rather due to cardiac autonomic dysfunction [33,34]. PD seems well to be significantly associated with increased left ventricular hypertrophy and diastolic dysfunction [34]. In the current case, ventricular dilatation and hypertrophy were present but seem rather related to manifest signs of chronic hypertension. In contrast, myocardial inflammatory reactions had been well-linked to gene-based COVID-19 vaccinations in numerous cases [9,35,36,37]. In one case, the spike protein of SARS-CoV-2 could also be demonstrated by immunohistochemistry in the heart of vaccinated individuals [28].

## 5. Conclusions

Numerous cases of encephalitis and encephalomyelitis have been reported in connection with the gene-based COVID-19 vaccines, with many being considered causally related to vaccination [31,38,39]. However, this is the first report to demonstrate the presence of the spike protein within the encephalitic lesions and to attribute it to vaccination rather than infection. These findings corroborate a causative role of the gene-based COVID-19 vaccines, and this diagnostic approach is relevant to potentially vaccine-induced damage to other organs as well.

## Figures and Tables

**Figure 1 vaccines-10-01651-f001:**
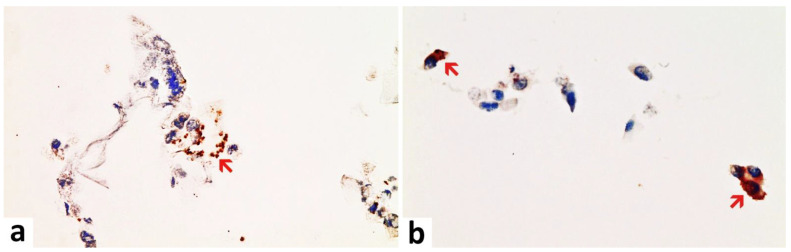
Nasal smear from a person with acute symptomatic SARS-CoV-2-infection (confirmed by PCR). Note the presence of ciliated epithelium. Immunohistochemistry for two SARS-CoV-2 antigens (spike and nucleocapsid protein) revealed a positive reaction for both as to be expected after infection. (**a**) Detection of the spike protein. Positive control for spike subunit 1 SARS-CoV-2 protein detection. Several ciliated epithelia of the nasal mucosa show brownish granular deposits of DAB (red arrow). Compared to nucleocapsid, the DAB-granules are fewer and less densely packed granular deposits of DAB. (**b**) Detection of nucleocapsid protein. Positive control for nucleocapsid SARS-CoV-2 protein detection. Several ciliated epithelia of the nasal mucosa show dense brownish granular deposits of DAB in immunohistochemistry (examples red arrows). Compared to spike detection, the granules of DAB are finer and more densely packed. Magnification: 400x.

**Figure 2 vaccines-10-01651-f002:**
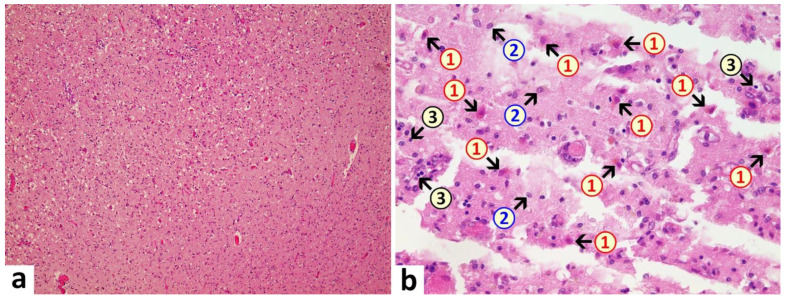
Frontal brain. Already in the overview image (**a**), prominent vacuolations with increased parenchymal cellularity are evident, indicative of degenerative and inflammatory processes. At higher magnification (**b**), acute brain damage is visible with diffuse and zonal neuronal and glial cell death, activation of microglia, and inflammatory infiltration by granulocytes and lymphocytes. 1: neuronal deaths (cells with red cytoplasm); 2: microglial proliferation; 3: lymphocytes. H&E stain. Magnification 40× (**a**) and 200× (**b**).

**Figure 3 vaccines-10-01651-f003:**
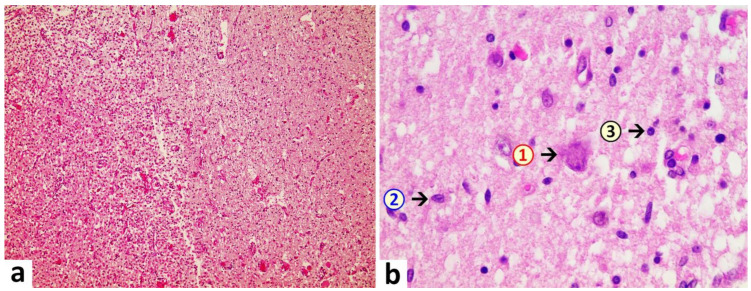
Brain, Nucleus ruber. In the overview image (**a**), note pronounced focal necrosis with increased cellularity, indicative of ongoing inflammation and glial reaction. At higher magnification (**b**), death of neuronal cells is evident and associated with an increased number of glial cells. Note activation of microglia and presence of inflammatory cell infiltrates, predominantly lymphocytic. 1: neuronal death with hypereosinophilia and destruction of cell nucleus with signs of karyolysis (nuclear content being distributed into the cytoplasm); 2: microglia (example); 3: lymphocyte (example). H&E stain. Magnification 40× (**a**) and 400× (**b**).

**Figure 4 vaccines-10-01651-f004:**
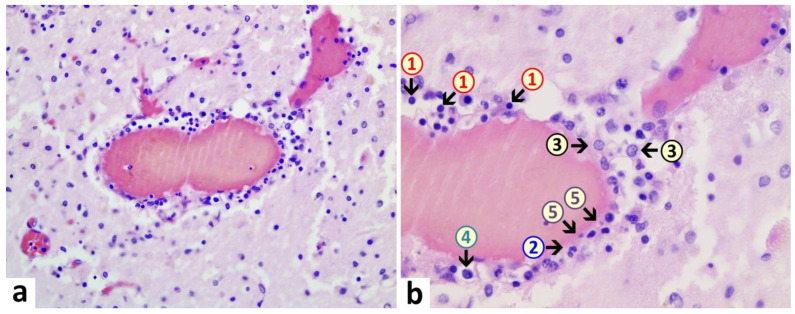
Brain, periventricular vasculitis. Cross section through a capillary vessel showing prominent signs of vasculitis. The endothelial cells (5) show swelling and vacuolation and are increased in number with enlargement of nuclei, indicative for activation. Furthermore, presence of mixed inflammatory cell infiltrates within the endothelial layer, consisting of lymphocytes (1), granulocytes (2), and histiocytes (4). The adjacent brain tissue also shows signs of inflammation (encephalitis) with presence of lymphocytes as well and activated microglia (3). H&E. Magnification: 200× (**a**) and 400× (**b**).

**Figure 5 vaccines-10-01651-f005:**
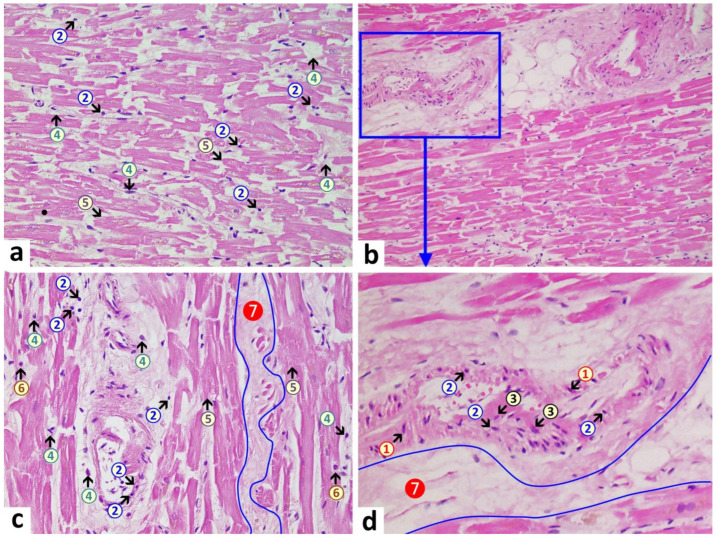
Heart left ventricle. (**a**): Mild lympho-histiocytic myocarditis.Pronounced interstitial edema (7) and mild lympho-histiocytic infiltrates (2 + 4). Signs of cardiomyocytic degeneration (5) with cytoplasmic hypereosinophilia and single contraction bands. (**d**): Arteriole with signs of acute degeneration and associated inflammation, associated by lymphocytic infiltrates (2) within the vascular wall, endothelial swelling and vacuolation (3), and vacuolation of vascular myocytes with signs of karyopyknosis (1). Within the vascular lumen (**d**), note plasma coagulation/fibrin clots adhering to the endothelial surface, indicative of endothelial damage. 1: pyknotic vascular myocytes, 2: lymphocytes, 3: swollen endothelial cells, 4: macrophages, 5: necrotic cardiomyocytes, 6: eosinophilic granulocytes, 7 (blue line): interstitial edema. H&E stain. Magnification: 200× (**a**) and (**c**), 40× (**b**), and detailed enlargement (**d**).

**Figure 6 vaccines-10-01651-f006:**
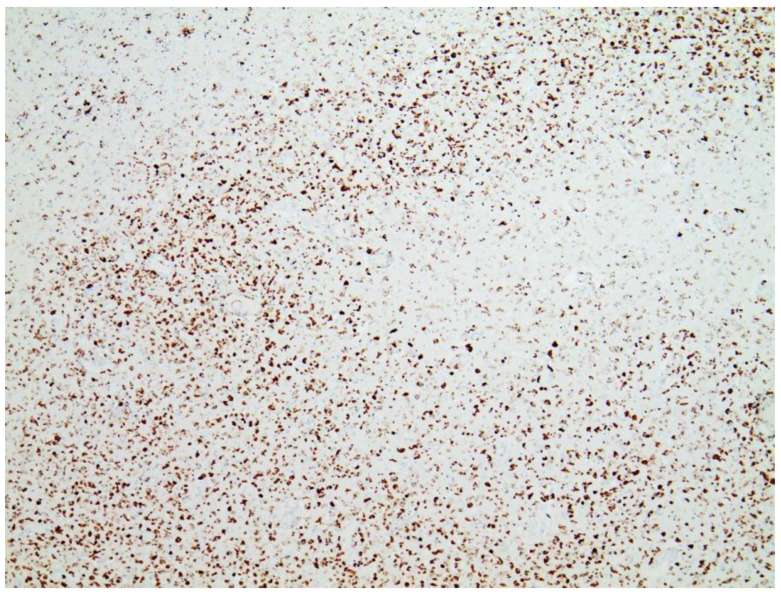
Frontal brain. Immunohistochemistry for CD68 (expressed by monocytic cells). Note map-like tissue destruction with the presence of CD68-positive microglial cells. Furthermore zonal activation of microglia (brown granules). Activation of the microglia means that tissue destruction has taken place in the brain, which is cleared/removed by macrophages (called microglia in the brain). Brown granules: macrophages/microglia. Magnification: 40×.

**Figure 7 vaccines-10-01651-f007:**
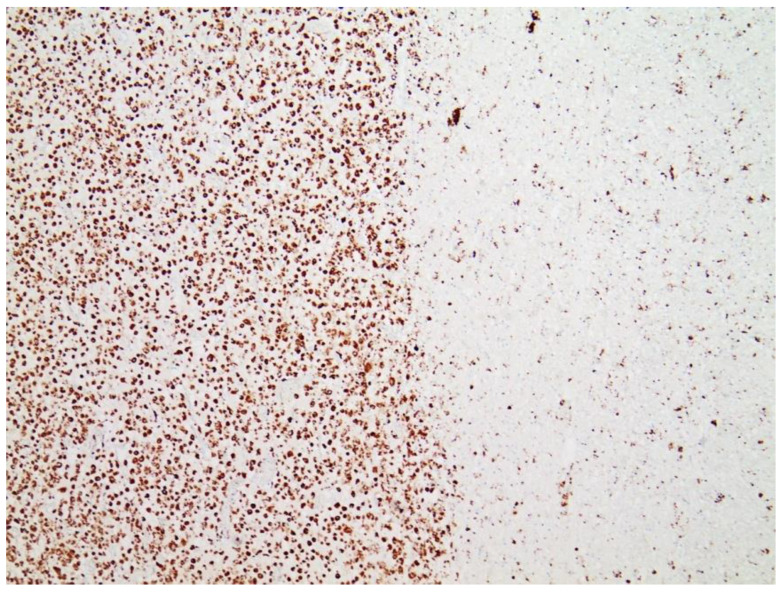
Brain. Nucleus ruber. Immunohistochemistry for CD68 (expressed by monocytic cells) shows abundant positive cells, indicative of zonal activation of microglia (brown granules). Magnification: 40×.

**Figure 8 vaccines-10-01651-f008:**
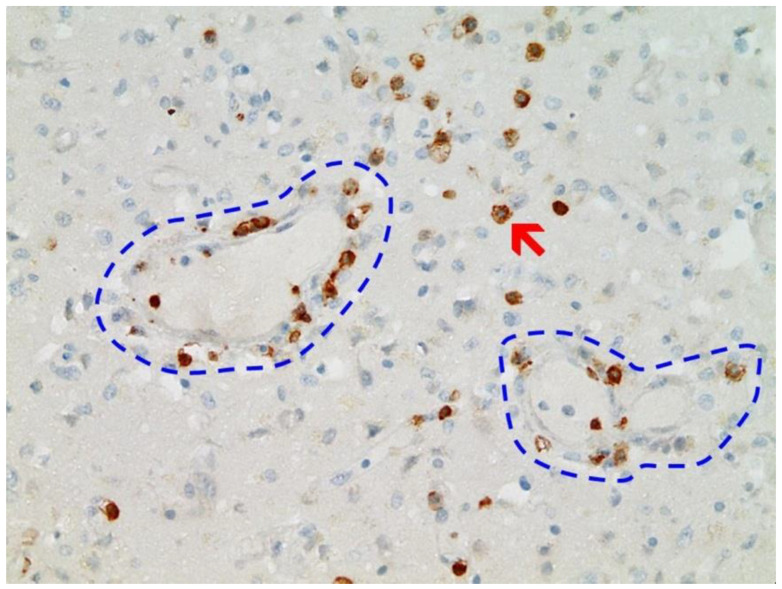
Frontal brain. Immunohistochemistry for CD3 (expressed by T-Lymphocytes) shows numerous CD3-positive lymphocytes (brown granules, red arrow highlights an example), particularly within the endothelium, but also in the brain tissue, indicative of lymphocytic vasculitis and encephalitis. Blue dotted lines: blood vessels. Magnification: 200×.

**Figure 9 vaccines-10-01651-f009:**
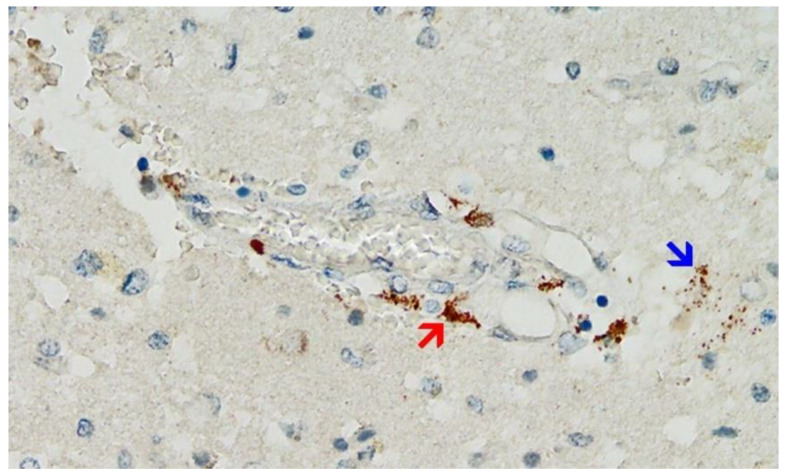
Frontal brain. Positive reaction for SARS-CoV-2 spike protein. Cross section through a capillary vessel (same vessel as shown in Figure 11, serial sections of 5 to 20 µm). Immunohistochemical reaction for SARS-CoV-2 spike subunit 1 detectable as brown granules in capillary endothelial cells (red arrow) and individual glial cells (blue arrow). Magnification: 200×.

**Figure 10 vaccines-10-01651-f010:**
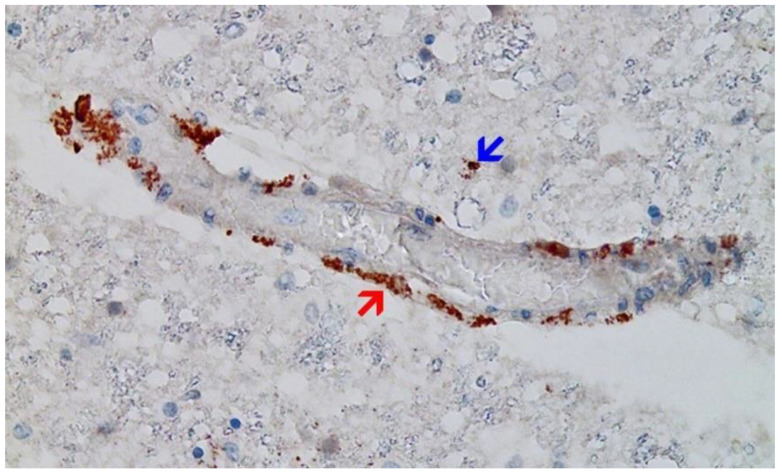
Brain, Nucleus ruber. The abundant presence of SARS-CoV-2 spike protein in swollen endothelium of a capillary vessel shows acute signs of inflammation with sparse mononuclear inflammatory cell infiltrates (same vessel as shown in Figure 12, serial sections of 5 to 20 µm). Immunohistochemical demonstration for SARS-CoV-2 spike protein subunit 1 visible as brown granules in capillary endothelial cells (red arrow) and individual glial cells (blue arrow). Magnification: 200×.

**Figure 11 vaccines-10-01651-f011:**
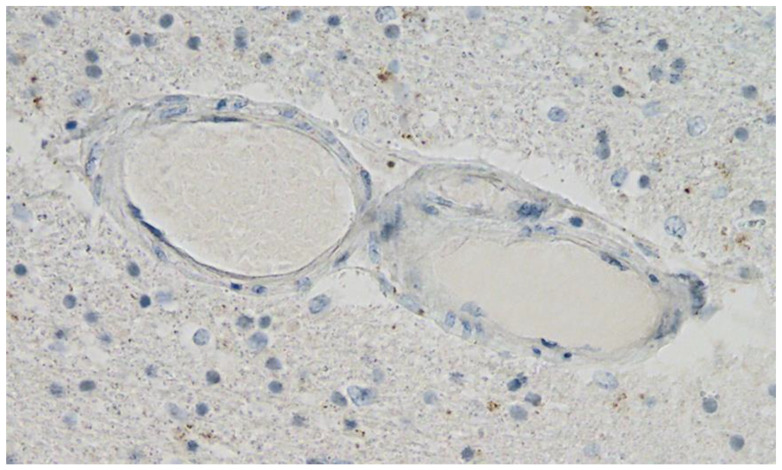
Frontal brain. Negative immunohistochemical reaction for SARS-CoV-2 nucleocapsid protein. Cross section through a capillary vessel (same vessel as shown in Figure 9, serial sections of 5 to 20 µm). Magnification: 200×.

**Figure 12 vaccines-10-01651-f012:**
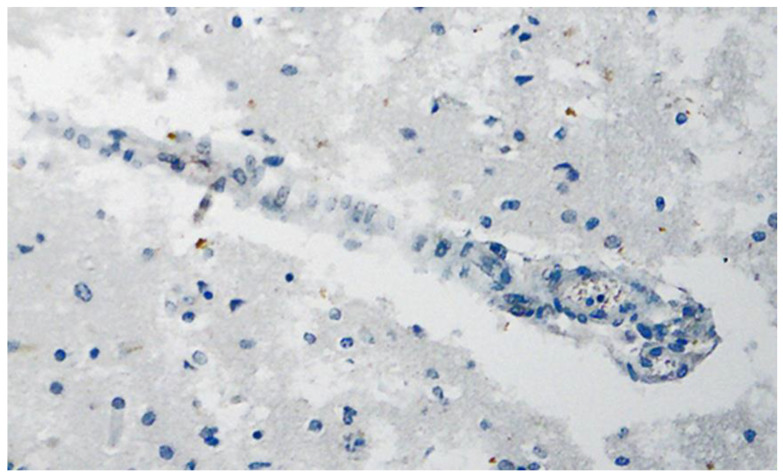
Brain, Nucleus ruber. Negative immunohistochemical reaction for SARS-CoV-2 nucleocapsid protein. Cross section through a capillary vessel (same vessel as shown in Figure 11, serial sections of 5 to 20 µm). Magnification: 200×.

**Figure 13 vaccines-10-01651-f013:**
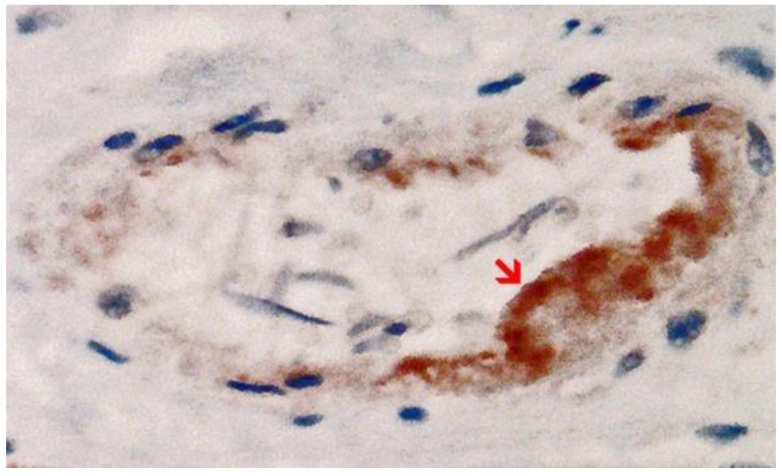
Heart left ventricle. Positive reaction for SARS-CoV-2 spike protein. Cross section through a capillary vessel (same vessel as shown in Figure 14, serial sections of 5 to 20 µm). Immunohistochemical demonstration of SARS-CoV-2 spike subunit 1 as brown granules. Note the abundant presence of spike protein in capillary endothelial cells (red arrow) associated with prominent endothelial swelling and the presence of a few mononuclear inflammatory cells. Magnification: 400×.

**Figure 14 vaccines-10-01651-f014:**
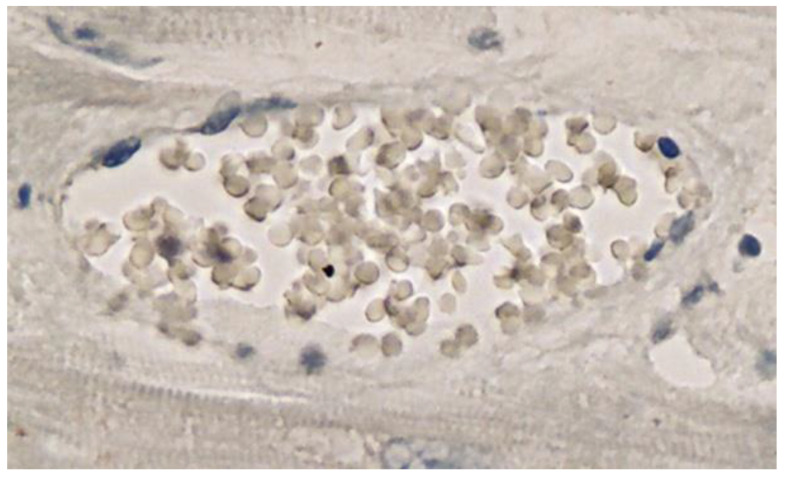
Heart left ventricle. Negative immunohistochemical reaction for SARS-CoV-2 nucleocapsid protein. Cross section through a capillary vessel (same vessel as shown in Figure 13, serial sections of 5 to 20 µm). Magnification: 400×.

**Table 1 vaccines-10-01651-t001:** Primary antibodies used for immunohistochemistry. Tissue sections were incubated 30 min with the antibody in question, diluted as stated in the table.

Target Antigen	Manufacturer	Clone	Dilution	Incubation Time
CD3 (expressed by T-Lymphocytes)	cytomed	ZM-45	1:200	30 min
CD68 (expressed by monocytic cells)	DAKO	PG-M1	1:100	30 min
SARS-CoV-2-Spike subunit 1	ProSci	9083	1:500	30 min
SARS-CoV-2-Nucleocapsid	ProSci	35–720	1:500	30 min

**Table 2 vaccines-10-01651-t002:** Anatomical Specifications.

Item	Measure
Body weight	60 kg
Hight	175 cm
Heart weight	410 g
Brain weight	1560 g
Liver weight	1500 g

## Data Availability

Data are available upon request.

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
