# Peer review of "A Case Report: Multifocal Necrotizing Encephalitis and Myocarditis after BNT162b2 mRNA Vaccination against COVID-19"

_vaccines, 2022, doi:10.3390/vaccines10101651_

Round 1
Reviewer 1 Report
This case report aims to represent a case of a 77-year-old man who was found to have neurological and cardiac inflammation upon autopsy. The man was vaccinated in May 2021 with the ChAdOx1 nCov- 19 vector vaccine, followed by two more doses with the BNT162b2 mRNA vaccine in July and December 2021, and died three weeks after receiving his third COVID-19 vaccination in January 2022.
Immunohistochemical staining revealed that the SARS-CoV-2 spike protein was evident in the tissues investigated. Since SARS-CoV-2 nucleocapsid-protein was not evidenced, the detected spike-protein was unrelated to a SARS-CoV-2 infection and the confirmed presence of the spike protein had to be attributed to the previous vaccination with the BNT162b2 mRNA vaccine.
Findings provide important evidence raising the attention to the possibility of serious side effects in the first week in patients receiving gene-based mRNA vaccines. Therefore, in clinical practice, cerebral and cardiac complications should be considered in patients receiving mRNA-based vaccines. Moreover, further well-designed preclinical studies are urgently required for future decision-making and health policy implementation.
The discussion is enough consistent with the evidence and
arguments presented.
The references result appropriate.
Line 76 “Table 1. Antibodies”: Explain better the table
Author Response
Comment: Table 1. Antibodies (line 76): Explain better the table
Response: Thank you for your attention. The title of table 1 has been changed.
Reviewer 2 Report
The introduction and discussion may be significantly shortened without detriment to the merit of the study.
The results of the autopsy should only include the detected abnormalities.
Author Response
Comment: The introduction and discussion may be significantly shortened without detriment to the merit of the study. The results of the autopsy should only include the detected abnormalities.
Response: Thank you for your time and effort to help meeting the required high standards of the journal for getting the manuscript published. The introduction and the discussion sections have been revised. As for the results of the autopsy, they are restricted to the essentials and did include already only abnormalities.
Reviewer 3 Report
The authors report a case of a 77-year-old man with Parkinson disease who died of aspiration pneumonia. He had been vaccinated against SARS-COV-2 infection with Ch AdOx1 nCovid 19, and with two doses of BNT162b2 mRNA vaccine. He died three weeks after the last jab. At autopsy, the authors found mild lymphocytic myocarditis and signs of multifocal necrotizing encephalitis. Because the authors found positive brain spike protein in the brain and in the myocardium, and negative nucleocapsid-protein in such organs, they claim that the disease is associated with vaccine-induced myocardial and brain lesions rather than active SARS-COV-2 infection. I have several concerns regarding this paper: 1) because there has been no concomitant myocardial necrosis, lymphocytic myocarditis seems not to be a suitable term. At most, the diagnosis should be done as mild borderline myocarditis; 2) lymphocytic infiltration is common in patients with chronic heart disease, mainly in the elderly; the presence of contraction bands suggests an ischemic origin for the lesions observed followed by lymphocytic infiltration. Therefore, I strongly suggest that the authors discuss this aspect in the Discussion section of the paper; 3) a reference should be provided emphasizing that such myocardial and cerebral lesions do not occur in patients with Parkinson disease; 4) references should also be provided to support the suggestion that the lesions observed (spike-protein positive, nucleocapsid-protein negative) in the tissues of concern, particularly the endotheliitis, are indicative of vaccine-induced endothelial lesions and not SARS-COV-2 infection; 5) line 380; this case report does not suggest “the possibility of serious side effects in the first week in patients receiving gene-based mRNA vaccines”, as the authors pointed out. Therefore, I suggest that the phrase be removed from the text. 6) line 384; I totally disagree with the statement “Moreover, further well-designed preclinical studies are urgently required for future decision-making and health policy implementation”. Massive vaccination has changed the clinical course of COVID-19. The complication reported here, if proved to be the consequence of vaccination, it will be unique. Even myocarditis is extremely rare, as the authors concede. Therefore, phase IV studies should be undertaken to detect complications of vaccination in the long-run. I suggest that the phrase be amended.
Author Response
Thank you for your attention and your time for a very detailed review. Your comments were very helpful and inspiring. As for the myocarditis, this is indeed only mild. There was no necrosis, that is right, but there were other signs indicative of acute myocardial alterations associated by inflammatory cell infiltrations (lympho-histiocytic). I added additional images to Figure 5, please see in the revised manuscript.
The most striking findings in the heart were the acute inflammatory lesions in the small myocardial blood vessels (as presented in figure 5). They consisted of endothelial swelling and vacuolation as well as vacuolation of vascular myocytes with signs of karyopyknosis. All this was associated by sparse lymphocytic infiltrates within the vascular wall. Furthermore, plasma coagulation/fibrin clots were noted adhering to the endothelial surface, indicative of endothelial damage. Important to note is, that in small vessels of the heart abundant presence of spike protein (but no nucleocapsid) of SARS-CoV-2 could be demonstrated within the endothelium (see Figure 13).
In addition to these vascular changes, there were mild early signs of degeneration and inflammation in the adjacent myocardium: hypereosinophilia and edema of cardiomyocytes with occasional karyopyknosis as well as some hypercontraction in single cardiomyocytes, associated with pronounced interstitial edema and mild presence of lympho-histiocytic infiltrates.
Hopefully, all this satisfies your queries and you can agree with the final diagnosis of “mild lympho-histiocytic myocarditis”. I certainly don’t want to say, that this had caused his death, but only wanted to express, that something was going on in the heart, which may have somehow contributed to his bad condition.
Comment 2: “Lymphocytic infiltration is common in patients with chronic heart disease, mainly in the elderly; the presence of contraction bands suggests an ischemic origin for the lesions observed followed by lymphocytic infiltration. Therefore, I strongly suggest that the authors discuss this aspect in the Discussion section of the paper.”
Response: Yes, I fully agree. It is quite difficult to distinguish chronic ageing-related changes in the heart from toxic lesions, in particular in the elderly with underlying chronic heart disease. Therefore, mild lympho-histiocytic infiltrations alone tell not much. In the current case, however, there were also the acute lesions in small cardiac blood vessels and also in the myocardium itself. All mild, I agree, but real. As for the presence of hypercontraction bands, it is to note that this was rather marginal in the presented case and nothing compared to what is known as “cardiac toxicity following catecholamine excess”. If this alone would have occurred, I would not even have mentioned it. The hypereosinophilia of cardiomyocytes, however, was clearly present. The same applies to the karyopyknosis of single cardiomyocytes. Furthermore, there was prominent interstitial edema and also edema of cardiomyocytes, accompanied by mild lympho-histiocytic infiltrations and acute vascular changes. All this together convinced me to call this a mild myocarditis. Please have a look on the additional images and on the discussion, where myocarditis had been addressed in particular.
Comment 3: “A reference should be provided emphasizing that such myocardial and cerebral lesions do not occur in patients with Parkinson's disease.”
Response: Thank you for this valuable remark. This had been addressed in the new discussion.
Comment 4: “References should also be provided to support the suggestion that the lesions observed (spike-protein positive, nucleocapsid-protein negative) in the tissues of concern, particularly the endotheliitis, are indicative of vaccine-induced endothelial lesions and not SARS-COV-2 infection.”
Response: This is a very important point and had been addressed in the new discussion.
Comment 5: “line 380; this case report does not suggest “the possibility of serious side effects in the first week in patients receiving gene-based mRNA vaccines”, as the authors pointed out. Therefore, I suggest that the phrase be removed from the text.”
Response: You are right, thanks for your comment. This was no more mentioned in the new discussion.
Comment 6: “line 384; I totally disagree with the statement “Moreover, further well-designed preclinical studies are urgently required for future decision-making and health policy implementation”.
Response: Thank you for this valuable remark. This topic of preclinical vaccine safety evaluation had been totally removed in the new discussion as it is beyond the scope of a case presentation.
Round 2
Reviewer 3 Report
I think that the paper has improved a lot with the author's review. I also think that it would be better to diagnose "heart involvement" instead of mild myocarditis. Because there is no myocyte necrosis, it would be better to diagnose borderline myocarditis. The other lesions in the adjacent myocardium, however, now very clear in the amended manuscript, might suggest the diagnosis of active mild myocarditis.